# Soft DAgger: Sample-Efficient Imitation Learning for Control of Soft Robots

**DOI:** 10.3390/s23198278

**Published:** 2023-10-06

**Authors:** Muhammad Sunny Nazeer, Cecilia Laschi, Egidio Falotico

**Affiliations:** 1The BioRobotics Institute, Scuola Superiore Sant’Anna, 56025 Pontedera, Italy; egidio.falotico@santannapisa.it; 2Department of Excellence in Robotics and AI, Scuola Superiore Sant’Anna, 56125 Pisa, Italy; 3Department of Mechanical Engineering, National University of Singapore, Singapore 117575, Singapore; mpeclc@nus.edu.sg

**Keywords:** DAgger algorithm, dynamic behavioral mapping, imitation learning, online optimization, Soft DAgger, soft robotics, soft robots control

## Abstract

This paper presents Soft DAgger, an efficient imitation learning-based approach for training control solutions for soft robots. To demonstrate the effectiveness of the proposed algorithm, we implement it on a two-module soft robotic arm involved in the task of writing letters in 3D space. Soft DAgger uses a dynamic behavioral map of the soft robot, which maps the robot’s task space to its actuation space. The map acts as a teacher and is responsible for predicting the optimal actions for the soft robot based on its previous state action history, expert demonstrations, and current position. This algorithm achieves generalization ability without depending on costly exploration techniques or reinforcement learning-based synthetic agents. We propose two variants of the control algorithm and demonstrate that good generalization capabilities and improved task reproducibility can be achieved, along with a consistent decrease in the optimization time and samples. Overall, Soft DAgger provides a practical control solution to perform complex tasks in fewer samples with soft robots. To the best of our knowledge, our study is an initial exploration of imitation learning with online optimization for soft robot control.

## 1. Introduction

The flexibility, deformability, and adaptability of soft robots offer significant potential to contribute to society by emulating the performance of biological systems in complex environments [1]. However, these advantageous features also pose challenges for accurate modeling and dynamic control of such platforms, especially in complex, unstructured, and unpredictable environments [2]. Although modeling soft robots is widely regarded as a challenging task, numerous researchers have proposed a variety of solutions [3]. These solutions span from geometrical [4,5], analytical [6,7], numerical [8,9], to learning-based approaches both in the kinematics and dynamics domain [10,11], respectively.

In the kinematics domain, several control strategies have been proposed for task-space trajectory following [12] including Jacobian estimation inspired by rigid robot controllers [13], fuzzy model-based controllers [14], Probabilistic Movement Primitives (ProMP)-based controller [15], and inverse kinematics-assisted control strategies [16]. In the dynamics domain, control approaches such as dynamics predictive control [17,18], open-loop [19] or closed-loop [20,21,22], learning-based control with a model [23,24], or in a model-free setting [10,25,26,27] have been proposed for a variety of tasks, including trajectory following, shape control, object/point tracking, and more.

However, practical real-life applications of soft robotics are yet to be fully realized. Carefully reflecting on the current literature, one may observe the dependency of the control solutions on the modeling approaches (analytical, numerical or data-driven) that may potentially be among the reasons to have affected said realization. While the literature acknowledges the potential drawbacks caused by constrained behavior captured by the modeling schemes and thereby limiting the capabilities of the control solutions, there is a paucity of work devoted to either deriving a control solution primarily on the physical robot or online optimizing the derived control solution on the soft robot. This way, it is also possible to limit the sim-to-real gap, which is recognized as one of the main sources of error for soft robot control [10].

It is generally agreed upon within the machine learning community that Imitation Learning (IL)-trained solutions often lack the necessary generalization capabilities (refer to Section 2.1). Various strategies have been proposed in the literature to enhance this generalization, with many relying on the incorporation of synthetic agents (often based on RL algorithms). While it is acknowledged that integrating such agents can indeed improve the generalization capability, they frequently exhibit significant sample inefficiency. This inefficiency stems from the agents’ reliance on exploratory actions within their environments in order to maximize their reward functions, which describe the desired task objectives. Consequently, substantial amounts of data samples may be required, and this requirement can increase dramatically with the complexity of the environment (e.g., involving virtually infinite degrees of freedom and inherent stochastic nature of soft robots) and the task at hand. Notable examples of this issue include an average of 3 million timetseps required for learning to follow a circular trajectory in [28], approximately 10 h of training time in [29] for a reaching task involving a humanoid, which eventually exhibits significant simulation-to-reality gap, and 200 episodes for learning to reach a single point in a 2D space with a da Vinci robot using a continuous on-policy IL algorithm [30]. Such solutions may not prove very practical for soft robots in particular, because of the magnitude of data samples required to perform a simple task.

In this study, we target this area and propose a sample efficient IL-based approach, which we refer to as Soft DAgger, to learn to perform a sophisticated task directly on a soft robotic arm. Soft DAgger does not require an in-depth understanding of the task or the soft robot; it only requires demonstrations of the desired task from a human and attempts to replicate them on the platform of choice. The primary contributions of this study include not only the presentation of a user-friendly and practical control strategy for soft robots, but also addressing the existing challenges in IL-based control (refer to Section 2). The contributions can be summarized as follows:Online control solution: Introduction of an algorithm for online control policy training and optimization using Transfer Learning (TL) and the Dynamic Behavioral Map (DBM) of a soft arm;Sample efficiency: A drastic decrease in the training time and samples required to learn to replicate desired expert demonstrations on a soft arm (almost 99.7% and 90% less samples and episodes required compared to an average Reinforcement Learning (RL) [28,31], and continuous IL algorithm [30], respectively);Robustness in the control solution: The ability to tackle excessive variability in expert task demonstrations due to the flexible shape, and reproducing the demonstrations on the soft arm while overcoming their inherent stochastic nature (an approximately 80% decrease in standard deviation of the Student Policy Reproduced Expert Demonstrations (SPREDs).

We wish to clarify that we are not suggesting that the only practical control solution is to learn on a physical soft robot, nor are we claiming that IL always yields an optimal control solution. However, we do propose that IL may offer a practical and sample-efficient solution for complex tasks involving soft robots. It is further acknowledged that the generalization capability of an IL-solution is still an active area of interest within the robotics community. Furthermore, expert demonstrations for soft robots may have significant variability due to their soft morphology, and they may not always be directly useful for policy learning, as soft robots experience varying kinematics when they come into contact with external objects. This issue is important if the expert task demonstrations involve the soft robots interacting with external objects. Addressing these challenges not only enhances the efficacy of this algorithmic approach for controlling soft robots, but also offers opportunities for future advancements in this domain.

## 2. Related Works

This section presents a literature review of the generic IL-related issues we intend to focus on in this study and the ways in which it is addressed in literature so far. Finally, we list the limited work conducted in IL for soft robot control.

### 2.1. IL and Generalization Capability

IL is an approach through which an expert teaches a robot to perform a task. An expert can be a human, a trained policy for a task, a synthetic agent, a demonstration from another robot or even a video demonstration of a desired task. The tasks successfully achieved with this class of algorithms extend from laboratory environment [32,33] to industrial applications [34,35]. Algorithms that learn based on exploring their workspace tend to requisite extensive resources (in terms of computational power and time) to produce a policy like RL. The policies thus trained have been found to be robust towards dynamic environments, as they are able to generalize over eluded states [36]. On the other hand, the salient attributes of IL include their ability to train a policy faster and without a handcrafted reward function [37]. In IL, exploration is not considered, so the policy is restricted to the knowledge taught by the expert. This causes the approach to fail under unfamiliar scenarios.

Zhang et al. [38] presented a novel approach to overcome this issue by introducing progressive learning, inspired by the way humans learn to perform a task from a few demonstrations. Their solution was tested with a 4-DoF experimental setup for pouring granular substance from one container to the other, and generalized across different backgrounds than the one with the expert demonstrations. Sasagawa et al. [34] presented a bilateral control strategy based on position and force. The demonstration setting included a human kinesthetically demonstrating the task on the master side while the robot followed the demonstrations on the slave side for a task of serving food on a plate using a spoon. The generalization capability of the proposed solution was tested against varying sizes of the served objects, the serving spoon length, and the height of the plate.

Among the proposed approaches to overcome said failures in IL, it is recommended in the literature to combine IL with RL. Preliminary results can be obtained with IL, and RL implementation can then be used to intelligently explore the environment to generalize over unfamiliar states as in Zhu et al. [39], Perico et al. [40], Sasaki et al. [41] and Stadie et al. [42]. In addition, IL complements RL in situations where RL cannot achieve pragmatic solutions, such as performing tasks in environments where the reward is either sparse or delayed [43]. While this combination has managed to successfully learn policies that are able to generalize well over various scenarios, the solutions presented by Zhu el al. and Perico et al. require more than 1 million and 120,000 (10 min long dataset at 200 Hz) time samples for policy training (approximately 99.75% and 98% more data samples than in Soft DAgger, respectively), which makes these solutions impractical for direct policy learning using soft robots.

On the other hand, the solution by Sasaki et al. has the capability to learn the policy from scratch using approximately 20,000 time samples, which is 87% more samples than in Soft DAgger. Conversely, the solution presented by Stadie et al. demonstrated the ability to generalize a previously trained policy in a new environment using approximately 2500 to 3000 time samples, thereby exhibiting similar sample efficiency as Soft DAgger. However, the solution by Stadie et al. was tested in simulation with a fixed Degrees of Freedom (Dof) experimental setup (i.e., three DoF or less), while soft robots are known to exhibit extremely non-linear behavior due to their virtually infinite degrees of freedom, so their solution, while it may be practical for learning to adapt even for soft robots (with increased number of iterations), may also require an impractical number of samples to train the first policy.

Most IL algorithms assume that the expert is an improved and final version of the supervisor. However, there may be scenarios in which the expert is also evolving with the passage of time, for example, a human also learning to perform the task while conducting the demonstrations, or a controller learning from a synthetic RL agent while also responsible for providing actions [44]. The training data may then include conflicting datasets for policy training, i.e., the same input data may lead to different actions depending on the supervisor at that particular instant. Balakrishna et al. [30] presented an approach to address this issue, which they claim outperforms deep RL baselines in continuous control tasks and drastically accelerates policy evaluation. Nevertheless, the issue of sample inefficiency persists due to the supervising RL-agent. In addition to RL, the concept of meta-IL (a class of algorithms that combine meta-learning with IL) has also been presented by Duan et al. [45] and Finn et al. [46], which enables a robot to learn more efficiently and generalize across multiple tasks in fewer iterations. It is acknowledged here that the solutions in [45,46] exhibit incredible sample efficiency for task generalization. However, the proposed solutions have been evaluated with considerably smaller state action space (compared to a soft robotic platform). Nevertheless, this class of algorithms can be an area to explore for soft robot control.

### 2.2. IL and Compounding Error

Although the above-mentioned approaches offer strategies to reduce the required number of iterations while introducing robustness, they do not address the common issue in IL of compounding error [47]. Additionally, many tasks require a sequential series of actions. Classical IL may not be suitable for such task settings, so an on-policy iterative class of algorithms under model-free behavior cloning was presented [48] (referred to as Data Aggregation or DAgger algorithms). These algorithms are guaranteed to produce deterministic policies under the sequential distribution of states [37], while also addressing the compounding error in IL. However, the incremental improvements in these algorithms require either human expert intervention [47,49] or synthetic agents [44] to correct each failed encounter in unfamiliar states. It is entirely possible that including human expert intervention to label or provide corrective actions for unfamiliar states may prove to be a sample efficient solution. However, it cannot be guaranteed that the human will always know the optimal action, nor can it be favorable, especially in field testing scenarios where safety is a concern. On the other hand, the RL-based synthetic agents can be sample inefficient as discussed in detail in the Section 2.1.

Since compounding error is a phenomenon that involves the policy performance degradation over time due to the policy error accumulation during the training phase, Soft DAgger mitigates this issue by introducing DBM. It is not only responsible for providing actions (refer to the Section 3.3) to reach a desired state as proposed by the expert, but also intervenes when an unfamiliar state is encountered. In a variant of Soft DAgger (as presented in Algorithm 1), the DBM is periodically improved for effective learning. The significance of this approach is highlighted in Section 4.2 for task learning.

Laskey et al. [50] addressed the challenge of compounding error without relying on human intervention or an iteratively learning agent. Their approach involves a policy referred to as the supervisor, where they introduce noise in desired task demonstrations. This aims to create a dataset that not only serves as a training sample, but also exposes the policy to unfamiliar states, allowing it recovery without compounding the error. This scheme has demonstrated effectiveness and sufficient sample efficiency against machine learning benchmarks. This is a promising way to tackle the compounding error; however, it may potentially lead to unsafe actions due to the noise, unlike Soft DAgger, where DBM is responsible for reducing the compounding error issue. On the other hand, Menda et al. [51] considers the same issue with a primary focus on the safety of the proposed actions. They incorporate an ensemble of policies responsible for exploration and error correction.
**Algorithm 1** Policy and Model Optimization (PMO)1:Input: $D_{B}$ and $D_{E}$                                                                   Stage 012:Output: $\tau_{\theta }$3:$f_{\phi_{0}}\left(\tau |X\right)\leftarrow D_{B}$                                                                       Stage 02 as per Equation (Equation 1)4:Initialize $\pi_{\theta }$, $D_{\pi }$, $L_{E}$5:**while** not done **do:**6:      Episode = sample($D_{E}$)7:      **for $i=0:L_{E}$ do:**8:             $\left(\mathrm{state}:x_{t_{c}},\;\mathrm{action}:f_{\phi_{i}}\left(x_{d}\right)\right)\leftarrow f_{\phi_{i}}\left(\tau_{\phi_{i}}|x_{d}\right)$9:             Append: $D_{i}\leftarrow \langle \;x_{t_{c}},\tau_{\phi_{i}}\;\rangle$10:           $D_{B}\leftarrow D_{B}\cup D_{i}$11:           $D_{\pi }\leftarrow D_{\pi }\cup D_{i}$12:           **Every** $n_{b}$   **and**   $D_{\pi }\geq 2\;L_{E}$ **do:**13:                 Compute loss for DBM: MSE ($L_{\phi }$)14:                 DBM Optimization: $f_{\phi_{i}}\leftarrow L_{\phi }$                             Stage 02 on repeat15:                 Compute Behavioral loss: ($L_{B}$)                           From Equations (Equation 3) and (Equation 4)16:                 Compute policy loss: ($L_{\theta }$)                                    From Equations (Equation 3) and (Equation 4)17:                 Policy Training & Optimization: $\pi_{\theta }\leftarrow L_{\theta }$        Stage 03

### 2.3. IL for Soft Robotics

Soft robots control using IL is an underexplored area. Among the limited literature found in this domain, the pioneering work is proposed by Malekzadeh et al. [32] where motion primitives learn from an octopus arm; for instance, the manner in which an octopus reaches towards or grasps an object is transferred to an octopus-inspired soft robot. Given the difference in the morphology of the robotic platform and the octopus arm, constant curvature constraints are applied to the motion primitives and the scheme is tested in a simulation environment. Beyond learning to behave like the biological system that inspired the robotic design, human surgical demonstrations have also been used to train a policy able to perform similar tasks with a surgical soft robot in a simulation environment, and with a 7-DoF rigid robot [33,52], respectively. While these solutions manage to learn and reproduce the demonstrations on the platform of choice in the simulation environment, there is no online learning or optimization involved and all the approaches require models of the underlying platforms. Additionally, the proposed solutions do not account for task repeatability, error accumulation or stochasticity in the soft robots. Soft DAgger, on the other hand, learns to perform the desired task directly on the soft robot and evaluates on the same soft robot as well without requiring its model. It also accounts for task repeatability by minimizing the standard deviation in the SPREDs (refer to the Section 4.2).

A related study is explored in [15], where the authors propose that specific skills can be acquired by deconstructing them into simpler motion primitives on a soft robotic arm. They define a series of sparse way-points and utilize the ProMP framework to plan qualitative trajectories between these points. Subsequently, an RL-based controller is trained to provide improvements in the run-time. Another approach is presented in [53], which employs the Central Pattern Generator (CPG) to generate oscillatory actuation values, resulting in periodic movements within the task-space of a soft arm. Then, RL is used to learn the CPG parameters to achieve desired trajectories. This technique establishes a dictionary of movement primitives, as also observed in [54], facilitating the replication of human demonstrations of periodic movements. The approach leverages an RL-based controller, specifically a continuous actor–critic learning automaton, initially trained in a simulation environment, and subsequently adapted and optimized on the soft robot using ProMP-based trajectories. While this work provides a novel approach to dynamically control soft robotic arms, it does not account for the sim2real gap associated with the RL-based controller that can lead to increased optimization time or degradation in policy performance.

Sequential decision-making capability, robustness, and task repeatability are desired from an IL-based policy for soft robot control, owing to the stochastic nature of the soft robots and significant variability present in the expert demonstrations. Additionally, the scarcity of meaningful training dataset necessitates sample efficiency. In numerous cases, soft robotic manipulators are pneumatically actuated, as is the case in our study [55,56]. While we can track the movements of a soft arm as demonstrated by an expert using tracking methods such as motion capture systems, we cannot record the corresponding pressure signals because the soft arm remains passive during expert demonstrations. These limitations are taken into consideration and addressed in our proposed solution (**Soft DAgger**).

## 3. Preliminaries

This sections describes the experimental setup, task description and requirements, and the proposed algorithm.

### 3.1. Experimental Setup

We utilized a two-module soft manipulator [55,56] as our testbed, consisting of three pairs of pneumatically actuated McKibben chambers in each module. The chambers were arranged at 120${}^{\circ }$ in the circular section of each module to enable bending in all directions. When each couple was individually actuated, it generated bending and extension, but when all couples were actuated simultaneously with equal pressure, the whole arm extension was achieved. The modules in the soft arm were independently actuated using six pneumatic control signals in total.

Nine markers were positioned on the robot to track the movement of both modules. Three markers were placed at the tip of the proximal module and three at the tip of the distal module. The remaining three markers were placed at the base of the proximal module to serve as an origin frame. All markers were arranged in a pattern resembling the vertices of an equilateral triangle. This allowed the center of the triangle serving as the tip position of that module. The experimental set was as shown in Figure 1.

The Vicon system, comprising eight Bonita cameras, was used to track all the markers. The cameras were set to capture different perspectives of the robot with redundancy. Since no single camera can capture all the markers at once, the full map of all markers was obtained by combining the images from all the cameras. The positions of the proximal and distal EE markers were recorded at a pre-selected frequency (100 Hz in this case) in terms of their x, y, and z components, and then imported into a Python-based workspace via ROS. The workspace was responsible for the DBM, and student policy training and optimization with feedback from the vicon system.

### 3.2. Task Description

The objective was to learn a policy capable of performing a desired task using a two-module soft robotic arm, as kinesthetically demonstrated by a human. Furthermore, we aimed to optimize the obtained results in real time to fit desired task requirements. To achieve this, we selected the task of writing letters in a 3D space within the task space of the soft arm, a task that has been demonstrated by various human experts. In the soft arm setup described in Section 3.1, 14 human experts were invited to demonstrate writing different letters, combinations of letters, and patterns in the 3D space by manipulating the soft arm. The movements of the tip of both modules of the soft arm were recorded during the expert task demonstrations. The diverse demonstrations of the same letter, provided by different human experts, were pooled together to form a comprehensive task dataset.

The demonstrations for the letter B are as shown in Figure 2a. Since the vicon system was set to record movements at 100 Hz, expert demonstrations were also recorded at 100 Hz. However, each episode (single expert demonstration) was sampled at 20 Hz from the vicon dataset to match the optimization loop (also running at 20 Hz). This resulted in approximately 250 task space states (12.5 s) for the letter B, 130 for letter M and Z (6.5 s each), and 450 for the word BRAIR and the pattern star (22.5 s each).

Based on the expert demonstrations, two key task requirements were taken into account. Firstly, the policy must imitate the expert demonstrations qualitatively, considering that these demonstrations might occasionally extend beyond the task space of the soft arm as also seen in Figure 2a. Secondly, the task should be repeatable with reduced variance. The latter task requirement is of particular importance in the context of soft robot control, as these robots inherently exhibit stochastic behavior due to the varying material properties.

### 3.3. Soft DAgger with DBM

Rusu et al. [44] employed policy distillation to train a control policy. This technique involves transferring knowledge of an extensive policy or expert to a smaller network or policy, known as the student policy. In [44], the expert is an RL-algorithm (Deep Q-networks or DQN)-based synthetic agent. The DQN-based expert not only trains a student policy using the distillation loss, but also undergoes reward maximization by means of self-improvement through exploratory actions (as illustrated in the Figure 3a). Such experts, though very robust for learning tasks from inaccurate environments, prove to be sample-inefficient, and may even lead to policy failing to converge due to conflicting state action pairs over time. In our approach, we substituted this agent with the DBM (Equation (Equation 1)) of the soft arm.
**Algorithm 2** Student Policy OpTimization (SPOT)1:Input: $D_{B}$ and $D_{E}$                                           Stage 012:Output: $\tau_{\theta }$3:$f_{\phi }\left(\tau |X\right)\leftarrow D_{B}$                                                   Stage 02 as per Equation (Equation 1)4:Initialize $\pi_{\theta }$, $D_{\pi }$, $L_{E}$5:**while** not done **do:**6:      Episode = sample($D_{E}$)7:      **for $i=0:L_{E}$ do:**8:             $\left(\mathrm{state}:x_{t_{c}},\;\mathrm{action}:f_{\phi }\left(x_{d}\right)\right)\leftarrow f_{\phi }\left(\tau_{\phi }|x_{d}\right)$9:             Append: $D_{i}\leftarrow \langle \;x_{t_{c}},\tau_{\phi }\;\rangle$10:           Policy Buffer: $D_{\pi }\leftarrow D_{\pi }\cup D_{i}$11:           **Every** $n_{b}$   **and**   $D_{\pi }\geq 2\;L_{E}$ **do:**12:                  Compute Loss: $L_{\theta }$                         From Equation (Equation 2)13:                  Policy Training: $\pi_{\theta }\;\leftarrow \;L_{\theta }$          Stage 03

#### 3.3.1. Dynamic Behavioral Map (DBM)

It is worth emphasizing that the teacher and student networks have access to training environments that vary in complexity and detail. In our case, the DBM served as the teacher. It mapped the task space of proximal and distal module of the two-module soft arm to its actuation-space at the maximum achievable velocity under a fixed operating rate (number of pneumatic signals per second). To train the DBM, we used an Artificial Neural Network (ANN) with a Long Short-Term Memory (LSTM) architecture. We gathered a dataset by running the soft robot through a series of random pressure signals, ranging from 0.1 to 1.2 bars, with the pressure signal dynamically saturating at randomly chosen ceilings in each chamber. This resulted in virtually non-repeating movements in the work space of the proximal and the distal module of the soft arm. We recorded the corresponding pressure and task space observations. The resulting dataset encompassed a duration of 5.0 min (300 s), with a total of 6000 observations recorded at a rate of 20 input signals per second. It is noteworthy that the dataset size required to train the DBM depends on several factors, including the desired accuracy, the complexity of the underlying platform, and the chosen mapping strategy (such as ANN, locally weighted projection regression, Gaussian process, etc.).

The ANN training was treated as a multi-variable time-series forecasting problem, with the goal of predicting the next action required to reach the desired task space state. The action ($\tau_{t}$) at a time instant *t* was predicted based on the current robot task space state ($x_{t}$), the previous four states ($x_{t-1}$, $x_{t-2}$, $x_{t-3}$, $x_{t-4}$), and the next state it was heading to ($x_{t+1}$), along with the corresponding actions ($\tau_{t-1}$, $\tau_{t-2}$, $\tau_{t-3}$, $\tau_{t-4}$, $\tau_{t-5}$), where x,τ∈R. Here, *x* corresponds to the 3D position of the tip of the proximal and distal module of the soft arm, and τ corresponds to pressure values for the six pneumatic chambers in the two-module soft arm. Including appropriate previous states as input in the DBM helped it generate unique output predictions for target states while taking into account the stochastic nature of the soft arm.

The DBM ($f_{\phi }$) predicted six outputs (equal to the number of pneumatic control chambers) with 66 input values. This input dimension was computed as per Equation (Equation 1). Both *x* and τ were mapped to a range from −1 to 1 before training. The training was performed using supervised learning with an ANN consisting of two hidden LSTM layers of 128 units each with a *Softsign* activation function, a dense output layer also with a *Softsign* activation function, and a 30% dropout layer before the output layer. The network was trained with a learning rate of 0.001, a batch size of 30, and 200 epochs, using the Adam optimizer. The hyperparameters and architecture were selected using a Bayesian optimization-based search algorithm provided by Keras [57].
(1)$$\begin{array}{c} f_{\phi }\left(\tau |X\right),\;\mathrm{where}\;X=\left(\{x_{i}{\}}_{i=t-4}^{t+1},\{\tau_{j}{\}}_{j=t-5}^{t-1}\right). \end{array}$$

#### 3.3.2. Soft DAgger

The Soft DAgger algorithm has three stages: offline data collection, online data collection, and a student policy training, as shown in the Figure 3b. In the first stage, two distinct datasets were gathered. The first dataset, called the motor-babbling dataset ($D_{B}=\left\{{\left(\tau_{t-1},\;x_{t}\right)}_{t=1},\;\ldots \right\}$ where t= 1:6000), was utilized to train the DBM ($f_{\phi }$), while the second dataset was specific to the task at hand. To obtain the task-specific dataset ($D_{E}$), 14 human experts were asked to demonstrate writing various letters (such as B, M, Z, S, and C), a combination of letters (such as BRAIR) and a pattern (such as star) several times in a 3D space by manipulating the soft robotic arm. Each character, word or pattern was considered as a single task here and each task had its own task-specific dataset buffer $D_{E}$ with the expert demonstrations. The states in the expert demonstrations were considered target states ($x_{d}$). A distribution was computed and plotted for each task as shown in the figures of Section 4.2 for the proximal and distal modules of the two-module soft arm in a viridis colormap (from colorcet library in matplotlib [58]). The blue colorband in the figures represents the variability in each distribution. Moreover, these distributions were normalized in the range from −1 to 1 based on their local minimum and maximum values.

In the second stage, first, the DBM was trained on the dataset sampled from the buffer $D_{B}$, and then it was used to predict actions ($\tau_{\phi }$) for states ($x_{d}$) sampled from the expert demonstrations. The current task space state of the soft arm ($x_{t_{c}}$) as a response to the predicted action $\tau_{\phi }$ was recorded in the instantaneous data buffer ($D_{i}$) where $D_{i}=\left\{{\left(x_{t_{c}},\;\tau_{\phi }\right)}_{t},\;\ldots \right\}$. The size of this buffer was equal to the episode size of the respective task. It is important to note here that the presence of the instantaneous buffer differentiates between the two variants of the Soft DAgger algorithm. In the first variant (Algorithm 2), there was no instantaneous buffer ($D_{i}$), and the DBM was trained only once using $D_{B}$, while for Algorithm 1, DBM underwent optimization periodically by sampling data from the joint buffer ($D_{B}$ + $D_{i}$).

In the third stage, instantaneous buffer was aggregated with the policy buffer $D_{\pi }$ and a student policy ($\pi_{\theta }$) was trained by minimizing the distillation loss as given in Equation (Equation 2) for Algorithm 2, and Equations (Equation 3) and (Equation 4) for Algorithm 1. The input to the student policy was the target state from the expert demonstration ($x_{d}$), current robot state ($x_{t_{c}}$), and the previous state ($x_{t_{c}-1}$), and the output was the pneumatic pressure ($\tau_{\theta }$) to reach the target state.
(2)$$L_{\theta }=\frac{1}{n_{b}}\sum_{i=0}^{n_{b}}{\left\|\tau_{\phi }-\tau_{\theta }\right\|}^{2},$$
(3)$$L_{\theta }=\frac{1}{n_{b}}\sum_{i=0}^{n_{b}}\alpha {\left\|\tau_{\phi }^{new}-\tau_{\theta }\right\|}^{2}+\left(1-\alpha \right)L_{B},$$
(4)$$L_{B}={\left\|\tau_{\phi }^{old}-\tau_{\theta }\right\|}^{2}.$$

Stages 01 and 03 remained the same for the two variants of Soft DAgger, but stage 02 changed. The first variant (SPOT—Algorithm 2) presupposed that the DBM was either an optimized mapping of the actual robot’s task space to actuation space or provided excellent generalization for unencountered states. By considering the DBM the ground truth, the student policy was optimized based on the loss function, as in Equation (Equation 2). In the second variant (PMO—Algorithm 1), the DBM and the student policy underwent periodic updates. The DBM was optimized using the Mean Squared Error (MSE) loss function by aggregating the instantaneous task-specific dataset ($D_{i}$) with dataset sampled from $D_{B}$. For student policy optimization, an additional loss (behavioral loss) was added to the distillation loss, as shown in Equations (Equation 3) and (Equation 4). A weight (α∈(0,1]) was assigned to the behavioral loss ($L_{B}$) to prevent drastic changes in the student policy updates while the DBM was undergoing updates as well. Therefore, if the Root Mean Squared Error (RMSE) between $\tau_{\phi }^{new}$ and $\tau_{\theta }$ was ≤$\tau_{max}$ over the $n_{b}$ batch, then α was set very close to one. This ensured the safety of the robot while allowing faster knowledge distillation from the DBM to the student policy. On the other hand, if RMSE between $\tau_{\phi }^{new}$ and $\tau_{\theta }$ was bigger than $\tau_{max}$, then α was set to proportionally less than one and the policy actions were trimmed (no value above $\tau_{max}$ was sent to the robot). In this case, proportional behavioral loss was taken into account during knowledge distillation while ensuring the safety of the robot. The pneumatic threshold ($\tau_{max}$) was set to 0.3 bars for the safety of the robot. Here, α sought to impose a trade-off between mimicking the expert and steady student policy training. The behavioral loss monitored the history of closeness of the student policy to the DBM while the DBM was undergoing improvement.

Saputra et al. [59] suggested a loss function that is similar to this. However, they utilized an MSE loss between the student’s predictions and ground truth values, as well as an imitation loss that utilized the MSE between the teacher’s and the student’s predictions. In our case, since the teacher network (DBM) was also serving as the ground truth value, our loss function was as shown in Equations (Equation 3) and (Equation 4) with a modified term called behavioral loss. Our loss combination was designed to prevent sudden changes in the student policy response because the DBM was also undergoing improvements unlike the one in [59], where there was ground truth available. Our approach, nonetheless, managed to achieve good results with appropriate sample efficiency for complex tasks.

## 4. Experimental Evaluation

This section focuses on the evaluation criteria for the student policy, and the results obtained from the two variants of Soft DAgger.

### 4.1. Student Policy Evaluation Criteria

Evaluating the improvements in results following each optimization iteration in this context is challenging. The use of quantitative metrics for evaluation such as MSE or Mean Absolute Error (MAE) is not suitable due to the high variability of the expert demonstrations, which can also extend beyond the operational space of the soft arm. Therefore, we use several qualitative indicators, derived from the distribution of the SPREDs, to evaluate the progress of the optimization process. These indicators encompass factors such as repeatability and consistency, pattern smoothness, reduction in dispersion, and mitigated presence of outliers. By considering these qualitative indicators, we can effectively evaluate whether the optimization process is improving the overall results.

Repeatability and consistency assesses the student policy’s ability to consistently reproduce the desired pattern multiple times with reduced variance, even when the demonstrations are drawn from a significantly diverse distribution of expert demonstrations. Pattern smoothness is evaluated by examining whether there are any abrupt deviations in the SPREDs compared to the expert demonstrations, and whether the SPREDs resemble the mean of the expert demonstrations’ distribution. Dispersion reduction, on the other hand, is evaluated by plotting a color band representing the 99% confidence interval around the mean across several test iterations after an optimization trial. The goal is for the dispersion (standard deviation) to decrease as the optimization trials progress. Finally, outlier reduction is evaluated by observing the decrease in the number of outliers over time. These outliers can accumulate errors and impact future predictions since current predictions provide feedback for the subsequent ones. The optimization loop in both variants continues until the desired letter becomes legible based on these criteria.

### 4.2. Results

All the figures in this section with scatter plots present three distinct yet related pieces of information. For instance, the left subfigure in Figure 4a illustrates data for the proximal module, while the one on the right shows data for the distal module of the soft arm. In both subfigures, the 2D scatter plots, along with the vertical colorbars, represent the operation range of each respective module. The colorbars at the top and bottom use the viridis and plasma colormaps to represent the means of expert demonstrations and SPREDs, respectively. The blue and black colorbands surrounding the means indicate the variability in expert demonstrations (as also in presented in Figure 2a) and the repeatability in the SPREDs, respectively. Finally, the standard deviation is calculated for each individual axis (x, y and z) of the SPREDs following each optimization trial. Subsequently, a norm is computed across these individual axis deviations. The norm is shown in all boxplots in this section (e.g., Figure 4f).

We apply Algorithm 1 to write the letters B, C, S, and a combination of letters (i.e., BRAIR). Subsequently, we apply Algorithm 2 for the letters M, Z, and a pattern (i.e., star). The final results for the tasks of writing C, S, BRAIR, M, and star are added in Appendix A and Appendix B for Algorithms 1 and 2, respectively. The reason behind using the two variants of Soft DAgger is discussed in Section 5.

#### 4.2.1. Policy and Model Optimization (PMO)

The scheme followed is outlined in Algorithm 1 (also shown in Figure 3b) for the desired task. The student policy and the DBM undergoes optimization in this approach. PMO takes the motor-babbling dataset buffer ($D_{B}$) and expert demonstrations buffer ($D_{E}$) as the input, and the output is a student policy ($\pi_{\theta }$) able to predict actions (pneumatic signals for all the chambers $\tau_{\theta }$) for writing the letter B. The DBM is trained offline using $D_{B}$ (refer to Section 3.3.1). A policy buffer ($D_{\pi }$), weights of the student policy network ($\pi_{\theta }$), and a variable $L_{E}$ (which is equal to the length of the episode sampled from $D_{E}$) are initialized.

The normalized desired task space states ($x_{d}$) are used as target states to predict an action by the current version of DBM ($\tau_{\phi_{i}}$). A temporary buffer ($D_{i}$) stores this action along with the associated task space state reached by the soft arm ($x_{t_{c}}$). After every batch ($n_{b}=\frac{1}{8}\;L_{E}$.), the temporary buffer combines the dataset with $D_{B}$ and $D_{\pi }$. The maximum size of these buffers is set to 2000 and $2L_{E}$ points, respectively. Both DBM and student policy undergo optimization based on the MSE, and Equations (Equation 3) and (Equation 4) do based on loss functions, respectively.

We use the preliminary version of the DBM for all the tasks performed with PMO. The student policy is periodically evaluated on the expert demonstrations that are not used during the policy training in Stage 03. Specifically, five expert demonstrations are sampled from the buffer $D_{E}$ and the current version of the student policy is executed with the soft robot with its different initial positions. A distribution is generated from the student policy executions (SPREDs). The mean and standard deviations of the distribution are plotted in a plasma colormap and a blue colorband, respectively, as shown in the Figure 4a–e, after executing 01, 03, 05, 07, and 08 full episodes of student policy optimization. The results for the additional tasks performed with PMO are shown in Figure A1 in Appendix A.

#### 4.2.2. Student Policy OpTimization (SPOT)

The scheme is outlined in Algorithm 2 and can also be visualized in Figure 3b without the instantaneous policy buffer $D_{i}$. The scheme was tested with the tasks of writing the letters M and Z. This approach also takes $D_{B}$ and $D_{E}$ as inputs, and outputs a student policy ($\pi_{\theta }$) able to predict actions (pneumatic signals for all the chambers $\tau_{\theta }$) for writing the desired letter. The scheme assumes the DBM predicts optimal actions to reach a target state within the operation range of the soft arm. The architecture, activation functions and other encompassing hyperparameters for the student policy are kept the same as in Section 4.2.1, except for $n_{b}$, which is changed to $\frac{1}{4}\;L_{E}$ because $L_{E}$ is smaller in these episodes. The policy ($\pi_{\theta }$) weights, $D_{\pi }$ and $L_{E}$ are initialized. Periodically (every $n_{b}$), the student policy weights are updated using the loss function ($L_{\theta }$ from Equation (Equation 2)) on the policy buffer ($D_{\pi }$).

The policy buffer stores $x_{t_{c}}$ and $\tau_{\phi }$. It is obvious that the student policy’s response is influenced by different versions of the DBM. Therefore, we divided this scheme into two categories. In the first category, we used the previously optimized DBM (from Section 4.2.1), while in the second category, we employed its preliminary version (offline trained only). In both categories, periodic student policy optimizations are conducted using independently selected episodes from $D_{E}$ for the letter Z. We observed that the results from both the categories differed, owing to the varying factor, i.e., the DBM. The final results for the first and second category, obtained after four student policy optimization trials, are shown in Figure 5a,c, along with the respective norms of their standard deviations in Figure 5b,d, respectively. The results for additional tasks performed with SPOT are shown in Figure A2 in Appendix B.

## 5. Discussion

Fourteen human experts were asked to manipulate the two-module soft robotic arm to demonstrate writing various letters, combination of letters, and patterns in the 3D space. The captured expert demonstrations exhibited significant variability (as shown in Figure 2a) due to factors such as robot’s soft morphology, absence of an initial reference point and the diverse writing styles of the human experts. Three distinct approaches for handling the soft arm, as observed from the human experts, are shown in Figure 2b–d. The distributions derived from these demonstrations are plotted (as shown in all the figures in Section 4.2) with blue colorband indicating the extent of this variability.

Here, we present the DBM of the robot as a key element in Soft DAgger, replacing the synthetic agents. This agent is sample efficient as it only requires 5 min of the dataset consisting of 6000 timesteps at a 20 Hz frequency, which results in approximately 95% fewer data samples for all the complex tasks performed in this study. Our solution successfully managed to learn to perform as complex a task as writing the character B using the soft arm in 100 s (as opposed to 10 h of training time for suboptimal outcomes in a reaching task). Each trial comprises approximately 250 timesteps for the character B and 130 timesteps for Z. With approaches such as Locally Weighted Projection Regression (LWPR) or a Gaussian process, the amount of dataset (and by that extension the training time) may further decrease even for a complex platform. In our current approach, it takes 8 trials to train a policy capable of writing the letter B and 4 trials for Z, despite the substantial variability in the expert demonstrations. We have observed an approximately 80% decrease in the standard deviation of the SPREDs for the letter B and a 70% decrease for Z.

The DBM is acting like a teacher/ground truth for the student policy training. Moreover, the consistent and approximate action prediction capability of this agent, based on the previous history, the expert-introduced variability, and varying initial positions of the soft arm collectively contribute to a sufficient generalization capability in the trained student policy.

By observing the decreasing black colorband around the mean (the plasma colormap) of the SPREDs across successive trials (in the Section 4.2), we can say that the results over successive trials are improving. Additionally, the outliers are controlled by introducing more previous states and actions as the input features in the DBM predictions. While the expert demonstrations may lie outside the operation range of the robot, normalizing these patterns according to their local minimum and maximum values reduces the chances of outliers, which also helps control the compounding error in the policy.

It is emphasized here that the student policy and the DBM have access to different training datasets. The DBM was trained on non-convex, non-repeating task space movements of the proximal and distal modules of the two-module soft arm, along with the associated random actuation signals. On the other hand, the student policy is trained on smaller datasets and has access only to task-specific data. This is the point where the distillation loss aids in obtaining a student policy that can predict as efficiently as the teacher. This underscores the significance of the DBM. We also acknowledge that the DBM may not provide the optimal set of actions the soft robot must take in order to reach a desired state, given the virtually infinite degrees of freedom of soft robots. Hence, incremental improvement to this agent might prove beneficial. Additionally, it was observed during the experiments with PMO that the task-specifc dataset (with periodic or cyclic movements) combined with the motor-babbling dataset appeared to optimize the DBM more effectively than a single extremely randomized actuation and resultant task space dataset.

This idea was further put to test during the training and validation of the student policy for the letter Z. We divided Algorithm 2 into two categories: one harnessing an optimized DBM, and the other employing its preliminary version. The preliminary version of the DBM was exclusively trained on the motor-babbling dataset. Consequently, the two categories of SPOT showed that the category that employs the improved DBM replicates expert demonstrations more rapidly and effectively compared to the second category within the same number of iterations, as seen from Figure 5a,c, respectively, after four iterations each. This insight indicates the underlying motivation for introducing the two variants of Soft DAgger (SPOT and PMO) that are presented in this work. In scenarios where we possess an optimal version of the DBM, Algorithm 2 is sufficient. However, it may not always be possible; Algorithm 1 is a more generic solution, producing a sample efficient control policy even for complex tasks such as writing a combination of letters (BRAIR as shown in the attached Appendix A and Appendix B), even when the demonstrations extend beyond the operation range of the soft arm.

Soft DAgger indeed demonstrated improved sample efficiency in task learning when compared to previous studies [39,40,41], offered a solution to overcome the compounding error problem without an RL-agent [15,44] or human intervention [47,49], and managed to reproduce significantly varying expert demonstrations on the platform of choice; the results, nonetheless, are only qualitative. A similar qualitative solution was also presented in [15], though with an RL-based policy acting as a teacher rather than a DBM of the soft arm. These qualitative results suggest that Soft DAgger is expected to perform well only if the task requirements include following the human demonstrations without explicitly treating them as the trajectory tracking problem with a desired accuracy. For enhanced accuracy in task execution within this setting, an additional exploration may also be required, which will introduce a trade-off between accuracy and sample efficiency.

## 6. Conclusions

The presented learning algorithm employs IL to swiftly train control policies for challenging tasks, as kinesthetically demonstrated by different human experts. Traditional IL-trained solutions often struggle with generalization, and the existing literature has proposed remedies often relying on either human intervention or the use of synthetic agents like RL-trained policies as “teachers” to address eluded states. Such solutions, though effective, tend to suffer from sample inefficiency, rendering them impractical for soft robotic applications. In contrast, our presented solution harnesses a DBM as the “teacher”. The DBM demonstrated improved sample efficiency, the ability to overcome compounding errors, and the capability to ensure task repeatability. It achieves this even with expert demonstrations that exhibit significant variation and extend beyond the operation range of the underlying soft arm. This strategy successfully trains a student policy capable of generalizing over the inherent stochastic nature of soft robotic motion, all without the need for RL agents or human intervention. However, it is crucial to point out that our approach is best suited for tasks that require reproducing the expert demonstrations qualitatively rather than precisely following a trajectory. Such task settings find potential applications in domains like elderly care or assistive soft manipulators mounted on kitchen counters or office desks, etc., where a limited number of demonstrations can enable the soft arm to either assist the elderly with activities like showering or help with cleaning the workspace, etc. Finally, our proposed approach not only offers a user-friendly and practical method for learning kinesthetically demonstrated tasks without the demonstrator having any in-depth knowledge of the task or the soft arm, but also has the potential to inspire future work in this domain.

## Figures and Tables

**Figure 1 sensors-23-08278-f001:**
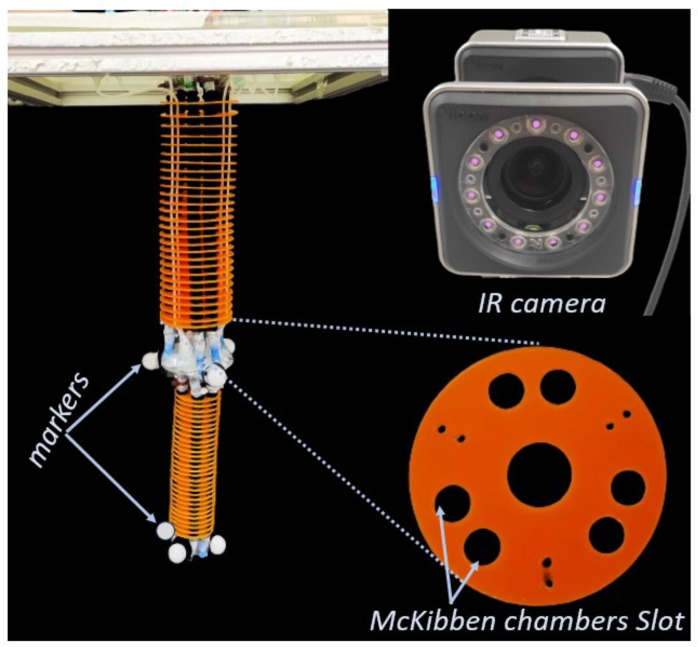
Two-module soft robotic arm with six pneumatic control signals placed in an eight-IR-camera-based motion capture system. The motion capture system is responsible for recording expert demonstrations. It also provides feedback about the tip movements of the soft arm during policy training using the markers placed on the soft arm.

**Figure 2 sensors-23-08278-f002:**
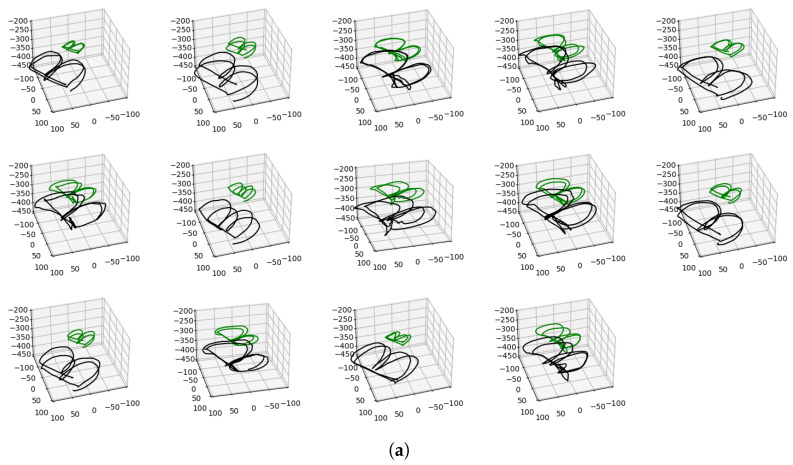
The variety in writing the letter B, as demonstrated by 14 experts, is illustrated in (**a**). The tip positions of the proximal and distal modules are represented in green and black, respectively. The experts exhibit distinct writing styles, as can be observed from variations in the size of the letter, the space covered, and the placement within the fixed scale of the axes. Figures (**b**–**d**) further demonstrate the distinct handling of the robot by the experts (as observed from the 14 experts) that contributes towards the diversity in the expert demonstrations. This figure is supported with Appendix A.

**Figure 3 sensors-23-08278-f003:**
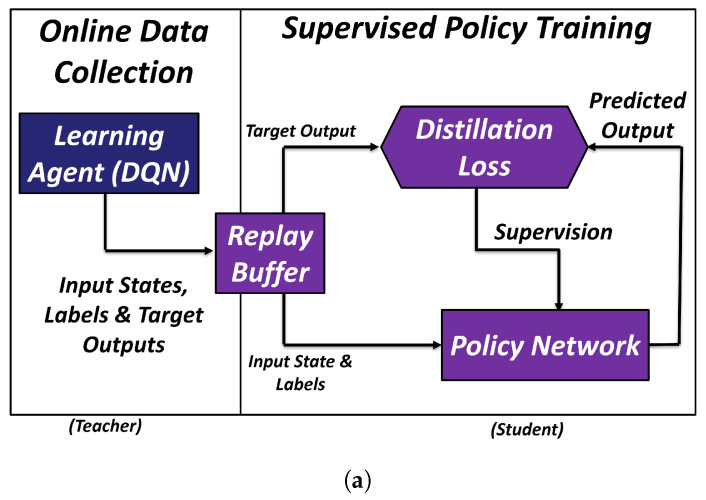
DQN is acting as the teacher here and is responsible for training a student network using policy distillation (**a**). It is also undergoing improvements by increasing its overall reward based on exploratory actions. In our approach, the DQN agent is replaced with the DBM as shown in Figure (**b**). It predicts actions for the expert demonstrations and acts as a supervisor for recovery from the eluded states. This scheme is a visualization of Algorithms 1 and 2 (without the instantaneous policy buffer $D_{i}$).

**Figure 4 sensors-23-08278-f004:**
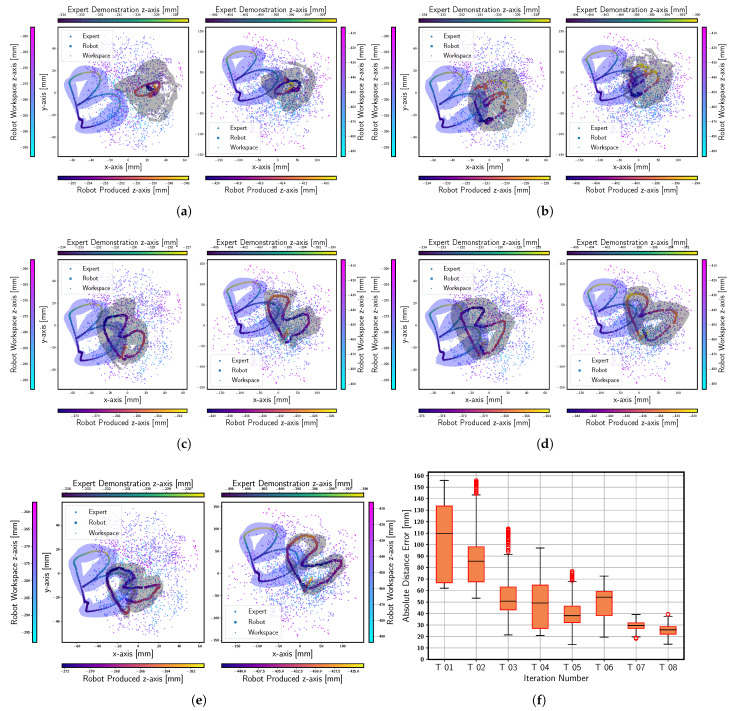
The distributions are plotted for the SPREDs after 01, 03, 05, 07 and 08 iterations of the student policy optimizations as shown in (**a**–**e**), respectively. Additionally, the standard deviations across all the SPREDs obtained after each student policy evaluation are summarized in (**f**).

**Figure 5 sensors-23-08278-f005:**
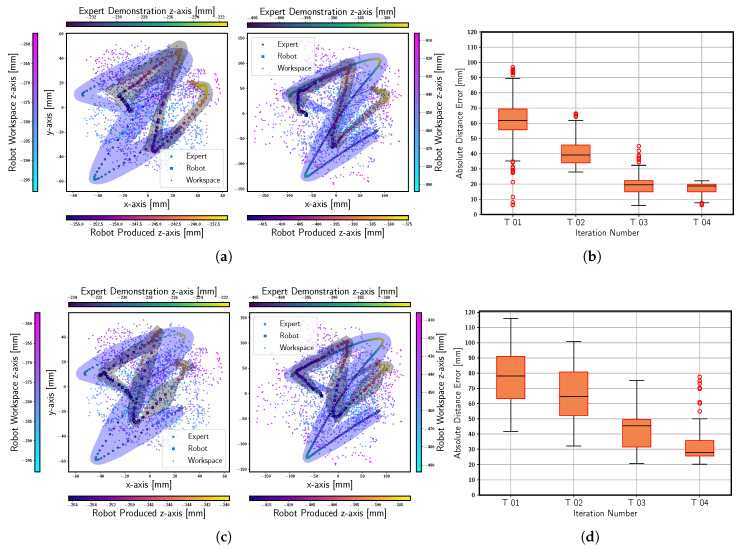
The categories of Algorithm 2 were employed to write the letter Z. The final outcomes for the first and second category of Algorithm 2 are presented in (**a**,**c**), respectively, after 04 iterations each. Additionally, the standard deviations across all the SPREDs obtained after each student policy evaluation are summarized in (**b**,**d**).

## Data Availability

Data is contained within the article and Appendix A.

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
