# Peer review of "Soft DAgger: Sample-Efficient Imitation Learning for Control of Soft Robots"

_sensors, 2023, doi:10.3390/s23198278_

Round 1

Reviewer 1 Report

please find it in attachment. 

Reviewer 2 Report

The article is a very relevant subject and written at a high level.

However, there are some points that need to be corrected.

The title of the article completely corresponds to it.

However, the article has a few points that need to be corrected:

- The authors should disclose in more detail the motivation for conducting this research. It is possible to demonstrate directions where such training may be needed in practice.

- Equation 3 should be divided into two equations.

- The conclusion should provide more accurate and compelling results with which the proposed model can reproduce a certain trajectory.
